# Molecular and structural basis of anti-DNA antibody specificity for pyrrolated proteins

Yusuke Anan[1], Masanori Itakura [1], Tatsuya Shimoda[1], Kosuke Yamaguchi[1], Peng Lu [2], Koji Nagata [2], Jinhua Dong[3,4], Hiroshi Ueda [3] & Koji Uchida [1,5✉]

Anti-DNA antibodies (Abs), serological hallmarks of systemic lupus erythematosus (SLE) and markers for diagnosis and disease activity, show a specificity for non-nucleic acid molecules, such as $N$-pyrrolated proteins (pyrP) containing $N^\varepsilon$-pyrrole-L-lysine (pyrK) residues. However, the detailed mechanism for the binding of anti-DNA Abs to pyrP remains unknown. In the present study, to gain structural insights into the dual-specificity of anti-DNA Abs, we used phage display to obtain DNA-binding, single-chain variable fragments (scFvs) from SLE-prone mice and found that they also cross-reacted with pyrP. It was revealed that a variable heavy chain (VH) domain is sufficient for the recognition of DNA/pyrP. Identification of an antigenic sequence containing pyrK in pyrP suggested that the presence of both pyrK and multiple acidic amino acid residues plays important roles in the electrostatic interactions with the Abs. X-ray crystallography and computer-predicted simulations of the pyrK-containing peptide-scFv complexes identified key residues of Abs involved in the interaction with the antigens. These data provide a mechanistic insight into the molecular basis of the dual-specificity of the anti-DNA Abs and provide a basis for therapeutic intervention against SLE.

[1] Laboratory of Food Chemistry and Life Science, Graduate School of Agricultural and Life Sciences, The University of Tokyo, Tokyo 113-8657, Japan.
[2] Laboratory of Food Biotechnology and Structural Biology, Graduate School of Agricultural and Life Sciences, The University of Tokyo, Tokyo 113-8657, Japan.
[3] Laboratory for Chemistry and Life Science, Institute of Innovative Research, Tokyo Institute of Technology, Yokohama 226-8503, Japan. [4] School of Rehabilitation Sciences and Engineering, University of Health and Rehabilitation Sciences, Qingdao, China. [5] Japan Agency for Medical Research and Development, CREST, Tokyo, Japan. ✉email: a-uchida@g.ecc.u-tokyo.ac.jp

Systemic lupus erythematosus (SLE) is a potentially fatal systemic autoimmune disease characterized by the increased production of autoantibodies (autoAbs). Of the multiple autoAbs reported in this disease, antibodies (Abs) directed against native DNA are known to be the most characteristic. The appearance of anti-DNA Abs in humans and murine models of lupus correlates with the progression of the disease. Compared to all other SLE-related autoAbs, anti-DNA Abs are considered to be the most pathogenic and have been implicated in the development of renal pathology. It has been suggested that the anti-DNA Abs are produced by B cells upon stimulation with specific antigens, such as DNA and the DNA-protein complexes[1]. These antigens are believed to be released into the blood from dead cells. In particular, undigested chromosomal DNA from apoptotic cells and DNA released as neutrophil extracellular traps are considered potential sources of extracellular DNA in patients with SLE[2]. However, due to the systemic features and complexity of this disease, it remains unclear what exactly are the primary stimuli that drive such autoAb responses and the mechanisms that regulate the overall pathological process of SLE.

Since the discovery of anti-DNA Abs, their specificity has been well studied. Anti-DNA Abs generally recognizes the phosphodiester backbone of DNA[3]. This means that electrostatic interactions may be involved in the Ab binding to DNA. Indeed, a comparison between a high number of anti-DNA Abs suggested that the majority of the Abs have at least one arginine residue in the complementary determining region 3 (CDR3) of the heavy chain, the region most critical for antigen recognition[4]. In addition, the importance of arginine residues for DNA recognition was demonstrated by reverse mutagenesis of an arginine residue to a germline amino acid[5]. These arginine residues are not derived from germline genes but rather from the non-templated nucleotide addition (N-addition) via VDJ recombination or somatic hypermutation in B cells. On the other hand, anti-DNA Abs are known to bind to various antigens other than DNA. As examples, anti-DNA Abs have been shown to recognize α-actinin[6], the N-methyl-d-aspartate (NMDA) receptor[7], and cardiolipin[8]. The binding of anti-DNA Abs to renal antigens, such as α-actinin, has also been proposed as an etiological mechanism of nephritis. More recently, N-pyrrolated proteins (pyrP) were identified as natural antigenic molecules with specificity for the anti-DNA Abs[9].

Protein N-pyrrolation is a covalent modification reaction that converts lysine residues to $N^\varepsilon$-pyrrole-L-lysine (pyrK) (Fig. 1a)[10]. We have previously shown[9] that (i) the immunization of BALB/c mice with the pyrrolated self-molecule accelerates the production of anti-DNA Abs, (ii) a spontaneous age-dependent increase in the IgG and IgM titers to DNA/pyrP is observed in the SLE-prone MRL-lpr mice, and (iii) SLE patients exhibit prominent increases in the Ab titers against both DNA and pyrP. Based on these findings, we proposed that protein N-pyrrolation is not simply a side effect associated with disease progression but may be the etiology of SLE and other autoimmune diseases. More recently, glycolaldehyde was identified as an endogenous source of the protein N-pyrrolation[9]. The presence of such endogenous pyrrolation factors indicates that the processes leading to the formation of pyrP can occur under physiological conditions. It has been proposed that the binding of pyrP for anti-DNA Abs may result from changes in the electrical properties of the proteins due to $N^\varepsilon$-pyrrolation of the lysine residues. However, the detailed mechanism of the binding of anti-DNA Abs to pyrP remains unknown. Therefore, to gain further insight into the specificity of the anti-DNA Abs for pyrP, this study used phage display, a biological and combinatorial discovery tool, to generate single-chain variable fragments (scFvs) and fully characterize their interactions with the antigens (DNA/pyrP).

## Results

### Specificity of anti-DNA scFvs for pyrP established from SLE-prone mice.

Purified spleen cells from SLE-prone MRL/lpr mice were used to generate a scFvs library in phage (Fig. 1b). The library size was estimated to be around $1 \times 10^8$, sufficient to cover the entire Ab repertoires of mice[11]. Polyclonal phage ELISA showed a successful selection of anti-DNA scFvs by biopanning (Fig. 1c). After two rounds of biopanning, 10 clones of scFv phage were randomly selected from the first screening and the other 10 clones of the scFv phage were randomly selected from the second screening, and the binding of these 20 clones to DNA was evaluated (Fig. 1d). Of the 12 clones, 8 clones (clones 3, 4, 6, 9, 13, 14, 17, and 18) consisted of full-length variable heavy (VH) and variable light (VL) regions, and 3 clones (clones 8, 12, and 16) contained internal amber stop codons (Amber codons 1–3). (Fig. 2). These eight clones consisting of the VH and VL regions were hereinafter referred to as scFvs DO1–DO8 (Fig. 3a). One remaining clone (clone 5) is described later.

A sequence analysis of the anti-DNA/pyrP scFvs using Ig BLAST and IMGT web resources revealed that, of the 8 scFvs established in this study, 5 scFvs (DO1, 3, 6, 7, and 8) were constituted by the V gene IGHV1-14*01, which is classified in the J558 family, and 1 scFv (DO2) was constituted by IGHV5-9*04, which is classified in the 7183 family. To assess the specificity of the anti-DNA scFvs, eight scFvs (DO1–DO8) were expressed and purified as hexa-histidine ($His_6$)-tagged recombinant proteins (scFv-$His_6$). The ELISA analysis showed that most of the anti-DNA scFvs showed a specificity to pyrBSA, despite their varying affinities for the antigens (Fig. 3b, c). These anti-DNA scFvs commonly recognized proteins (serum albumins, transferrin, and IgG) that had been pyrrolated with 1,4-butanedial (Supplementary Fig. 1). Moreover, among the tested aldehydes, the anti-DNA scFvs cross-reacted almost exclusively with pyrP prepared by the incubation of BSA with 1,4-butanedial (Supplementary Fig. 2).

### VH domain alone is sufficient for the recognition of DNA/pyrP.

On the other hand, anti-DNA scFvs screening unexpectedly established a clone consisting of a truncated scFv (VH-FR4) with a VH domain and a VL region (FR4) present (Figs. 2, 4a). This result suggested that the VH domain, but not the VL domain, may be sufficient for antigen (DNA/pyrP) recognition. To test this hypothesis, two recombinant proteins, a VH with a frame region of a light chain (VH-FR4) and a VH domain only (VH), were expressed and purified from clone 5 as $His_6$-tagged recombinant proteins (Fig. 4b). The ELISA analysis using these domain Abs from clone 5 revealed that not only VH-FR4 but also the single VH domain can cross-react with DNA/pyrP (Fig. 4c, d). To further validate the potential recognition of antigens by the VH domain alone, the VH and VL domains were separately prepared from other scFvs, such as VH-$His_6$ and VL-$His_6$, respectively. Since the amino acid sequences of the VL domains of DO2 and DO3 were identical, they were created as a single VL-$His_6$ (DO2/DO3 VL). VH DO8 could not be expressed for unknown reasons. Figure 5a showed that both VH DO1 and DO2 cross-reacted with DNA/pyrP whereas VH DO7 was rather specific to dsDNA. VH DO5 showed non-specificity to all three antigens, and the remaining VH Abs (DO3, 4, and 6) did not show significant binding. VL-$His_6$ hardly recognized the antigens (Supplementary Fig. 3). A competitive ELISA using model DNA showed that the domain VH Abs (DO1 and DO2) is relatively specific for guanine-containing sequences compared to the other

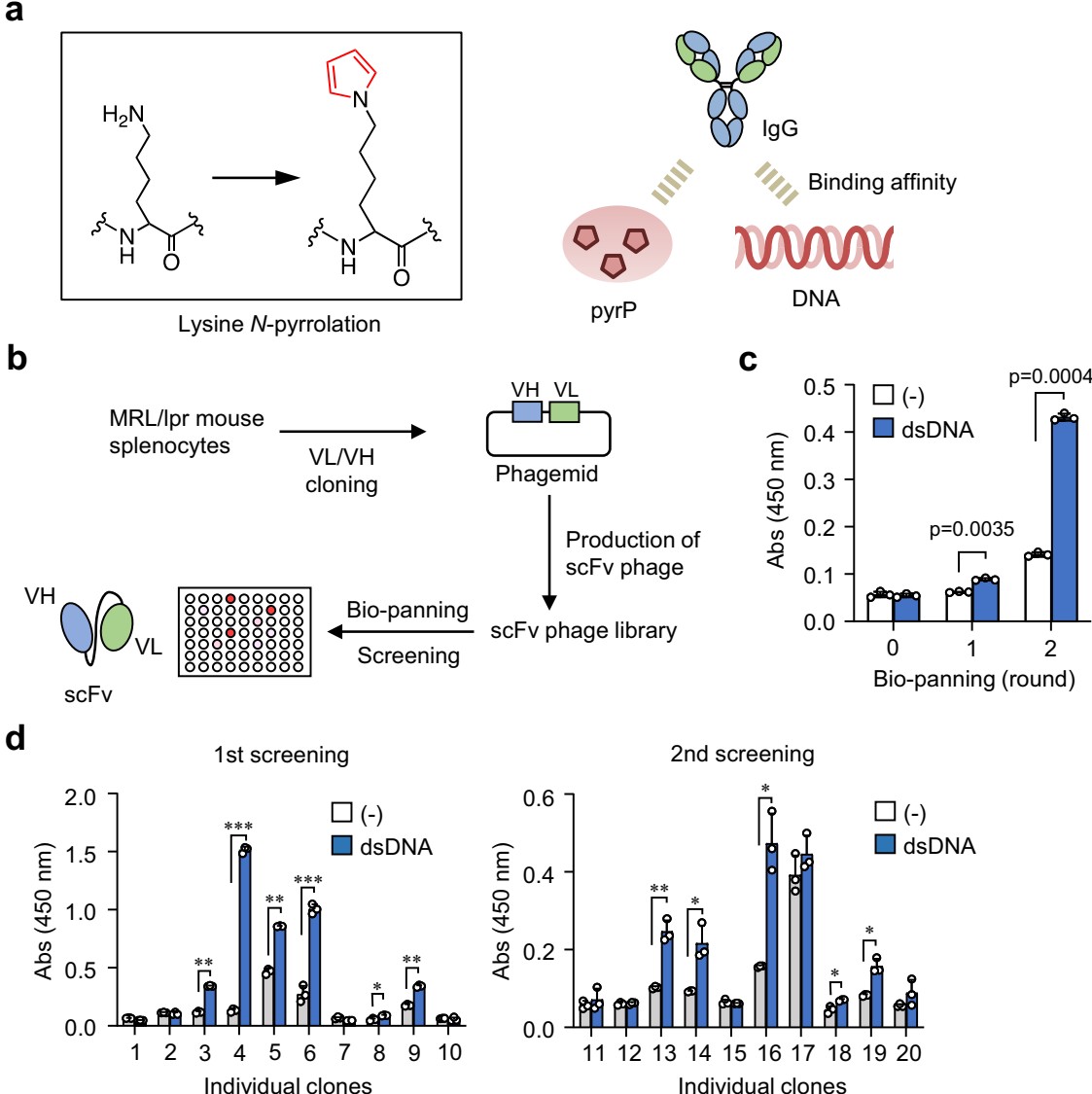

**Fig. 1 Establishment of anti-DNA scFvs. a** Schematic representation of lysine $N^\epsilon$-pyrrolation in proteins (*left*) and dual-specificity of anti-DNA Abs for DNA and pyrP (*right*). **b** Scheme for the establishment of anti-DNA scFvs from SLE-prone mice. **c**, Bio-panning selection of anti-DNA scFvs. The selection was evaluated by ELISA against sermon sperm genomic DNA using $1 \times 10^{10}$ pfu polyclonal phage. Data are mean ± S.D. of triplicate samples (representative of three independent experiments). Student's *t*-test (two-sided). **d** Screening of anti-DNA scFv. The binding capacity to DNA was evaluated by ELISA using monoclonal scFv-G3P from the bacterial cultures of individual clones. Screenings (*left*, first screening; *right*, second screening) were performed twice with 10 clones each. Data are mean ± S.D. of triplicate samples (representative of three independent experiments). Student's *t*-test (two-sided), *$p < 0.05$; **$p < 0.01$; ***$p < 0.001$.

sequences (Fig. 5b). We have also tried to measure the affinity of an antibody to pyrP and DNA using SPR. So far, we have obtained data on VH Abs (DO1 and DO2) binding to pyrP (Supplementary Fig. 4), showing the binding of pyrP to the VH Abs. Thus, we established that some VH domains, at least VH DO1 and DO2, could recognize the antigens (DNA/pyrP) without VL domains.

**Arginine residues in the VH domain Abs are involved in the recognition of DNA/pyrP.** Previous studies have shown that arginine residues play an important role in the recognition of DNA by anti-DNA Abs[5,12]. Consistent with this finding, the VH domains of the anti-DNA scFvs contain multiple arginine residues, mainly in CDR3 (Fig. 2). These findings, along with the fact that lysine *N*-pyrrolation confers an electronegativity and electronic properties to proteins that are electromimics of DNA[9],

suggest the involvement of electrostatic interactions between the positively charged arginine residues on the VH domains and negatively charged DNA/pyrP. To test this hypothesis, we examined the effect of the ionic strength on the binding of VH domains to antigens. The two domain Abs, VH DO1 and VH DO2, showing the greatest binding among the seven clones for DNA/pyrP, were incubated with either pyrBSA or DNA in a phosphate buffer containing various concentrations of NaCl. As shown in Fig. 6a, the binding activities of VH DO1 and VH DO2 toward pyrP and DNA were significantly reduced by NaCl in a concentration-dependent manner. We also carried out an experiment regarding the effect of serum addition on the antibody binding to the ligands (pyrBSA and dsDNA). The data showed that the serum inhibited the antibody binding to the pyrBSA and dsDNA in a dose-dependent manner (Supplementary Fig. 5). The data also support our hypothesis that the binding

| Position | | 26 | 33 | | 52a | 56 | | 95 | 100a b c d e f g h 102 |
|---|---|---|---|---|---|---|---|---|---|
| Clone | FR1 | HCDR1 | | FR2 | HCDR2 | | FR3 | | HCDR3 |
| scFv DO1 | · · · | K A S G Y T F T S Y V M H W | · · · I G Y | I N P H N D G T K Y N | · · · | Y Y C A R K L **R** G F - - - - - - - - A Y |
| scFv DO2 | · · · | A A S G F T F S S Y T M S W | · · · V A T | I S S G G G **R** T Y Y P | · · · | Y Y C V R Q **R** G G **R** G Y A M - - - - D Y |
| scFv DO3 | · · · | K A S G Y T F T S Y V M H W | · · · I G Y | I N P Y N D G T K Y N | · · | Y Y C V V H Y T G H Y Y A M - - - - D Y |
| scFv DO4 | · · · | K A S G Y T F T **R** Y W I N W | · · · I G N | I Y P G S S **R** T N Y N | · · · | Y Y C A R G D **R** S G S **R** Y Y Y T M - - D Y |
| scFv DO5 | · · · | K A S G Y T F T S Y W I N W | · · · I G N | I Y P G S S S T N Y N | · · · | Y Y C A R **R** G L **R R** P Y W Y F - - - - D V |
| scFv DO6 | · · · | K A S G Y T F T S Y V M H W | · · · I G Y | I N P Y N D G T K Y N | · · · | Y Y C A R E G P **R R** Y Y Y A M - - - - D Y |
| scFv DO7 | · · · | K A S G Y T F T S Y V M H W | · · · I G Y | I N P Y N D G T K Y N | · · · | Y Y C A R **R** P P H Y Y **R** Y D F N Y A M D Y |
| scFv DO8 | · · · | K T S G Y T F T N Y V M H W | · · · I G Y | I N P N D V A K Y N | · · · | Y Y C A R S **R** S S G F - - - - - - - - A Y |
| Amber codon 1 | · · · | K A S G Y T F T S Y G I S W | · · · I G E | I Y P **R** S G N T Y Y N | · · · | Y F C A R **R** P F - - - - - - - - - - A Y |
| Amber codon 2 | · · · | K A F G Y T F T N H H I N W | · · · I G Y | I N P Y N D Y T S Y N | · · · | Y Y C A R S **R** Q L G L **R** W F - - - - - A Y |
| Amber codon 3 | · · · | K A S G Y T F T S Y G I S W | · · · I G E | I Y P **R** S G N T Y Y N | · · · | Y F C A R **R R R** I F Y Y F - - - - - - D Y |
| VH-FR4 | · · · | K A F G Y T F T N H H I N W | · · · I G Y | I N P Y N D Y T N Y N | · · · | Y Y C A R **R** G Y A M - - - - - - - - D Y |

| Position | | 24 | 30a b c d e | 34 | | 50 | 56 | | 89 | 97 |
|---|---|---|---|---|---|---|---|---|---|---|
| Clone | FR1 | LCDR1 | | | FR2 | LCDR2 | | FR3 | | LCDR3 |
| scFv DO1 | · · · I S C | R S S Q S L L H S N R N T Y L H W Y L | · · · | L I Y | K V S N R F S G V P | · · · | Y F C S Q S T H V P L T |
| scFv DO2 | · · · I S C | R S S Q S L V H S N R N T Y L H W Y L | · · · | L I Y | K V S N R F S G V P | · · · | Y F C S Q S T H V P L T |
| scFv DO3 | · · · I S C | R S S Q S L V H S N R N T Y L H W Y L | · · · | L I Y | K V S N R F S G V P | · · · | Y F C S Q S T H V P L T |
| scFv DO4 | · · · I S C | R S S Q S L V H S N G N T Y L H W Y L | · · · | L I Y | K V S N R F S G V P | · · · | Y F C S Q S T H V P - T |
| scFv DO5 | · · · · M T C | R A S E N I D - - - - - S N L V W Y Q | · · · | L V Y | T A T N L A D G V P | · · · | Y Y C Q H F Y G T P R T |
| scFv DO6 | · · · I S C | R A S K S V S T S S Y - S Y M H W Y Q | · · · | L I K | Y A S Y L E S G V P | · · · | Y F C Q H S R E F P R T |
| scFv DO7 | · · · I T C | R S S Q S L V H S N G N T Y L H W Y L | · · · | L I Y | K V S N R F S G V P | · · · | Y Y C S Q S T H V P L T |
| scFv DO8 | · · · I S C | R S S Q S L V H S N G N T Y L H W Y L | · · · | L I Y | K V S N R F S G V P | · · · | Y F C S Q S T H V P - T |
| Amber codon 1 | · · · I S C | R S S Q S L V H S N G N T Y L H W Y L | · · · | L I Y | K V S N R F S G V P | · · · | Y F C S Q S T H V P L T |
| Amber codon 2 | · · · I S C | S A S S S V S S - - - - S Y L Y W Y Q | · · · | W I Y | S T S N L A S G V P | · · · | Y Y C Q Q Y D S S P F T |
| Amber codon 3 | · · · I S C | R S S Q S L V H S N R N T Y L H W Y L | · · · | L I Y | K V S N R F S G V P | · · · | Y F C S Q S T H V P L T |

**Fig. 2 Amino acid sequences of VH and VL domain of scFv clones.** The amino acid sequences of the CDRs of the VH domain (*Upper*) and the VL domain (*Lower*) are shown. The frame region (FR) sequences are abbreviated by dots, the alignment-derived spaces are represented by a hyphen, and the arginine residues in HCDR are colored blue. The amino acid positions, CDRs, and FRs were determined by the Chothia numbering system.

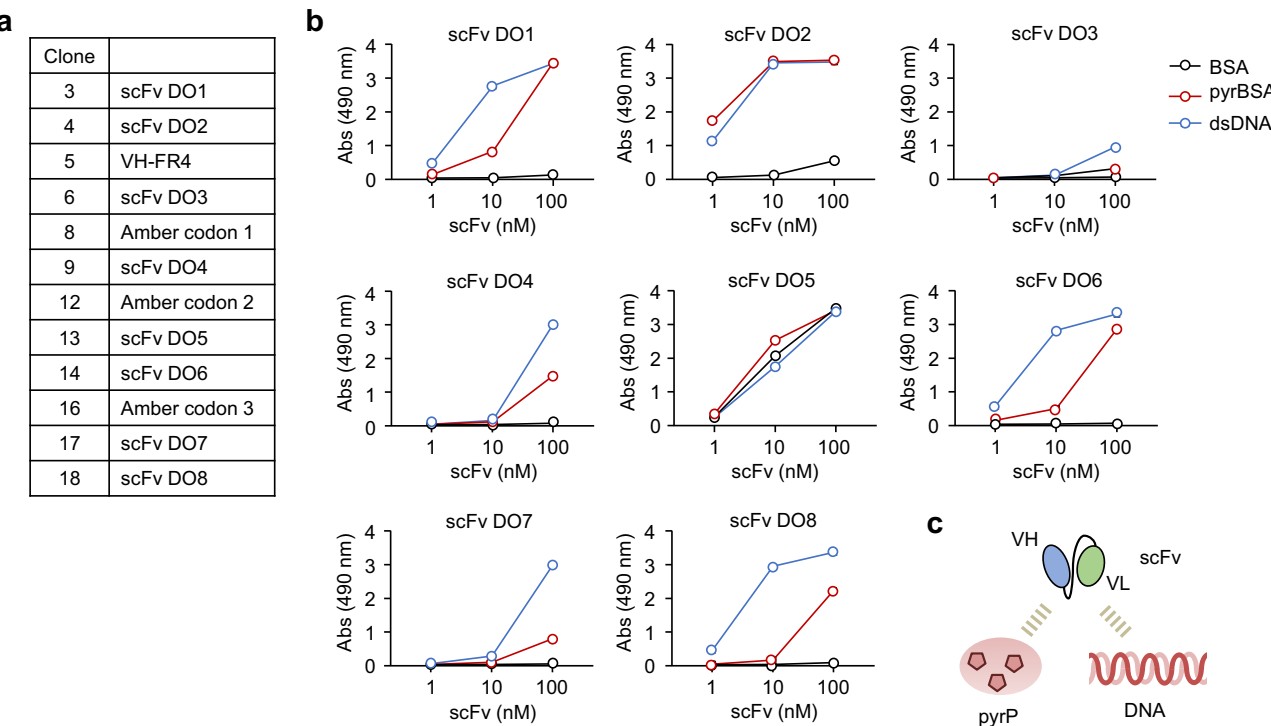

**a**

| Clone | |
|---|---|
| 3 | scFv DO1 |
| 4 | scFv DO2 |
| 5 | VH-FR4 |
| 6 | scFv DO3 |
| 8 | Amber codon 1 |
| 9 | scFv DO4 |
| 12 | Amber codon 2 |
| 13 | scFv DO5 |
| 14 | scFv DO6 |
| 16 | Amber codon 3 |
| 17 | scFv DO7 |
| 18 | scFv DO8 |

**Fig. 3 Dual-specificity of anti-DNA scFvs for pyrP. a** List of scFv-clones established in Fig. 1d. **b** Binding of scFv DO1–DO8 to antigens (BSA, pyrBSA, and DNA). Antigens (1 µg/well) immobilized on the ELISA plate were incubated with monoclonal scFv-His$_6$ at concentrations of 1, 10, and 100 nM. Binding was detected using the HRP-conjugated anti-His$_6$-tag antibody. Data are mean ± S.D. of triplicate samples (representative of three independent experiments). **c** Schematic representation of dual-specificity of anti-DNA scFvs for pyrP.

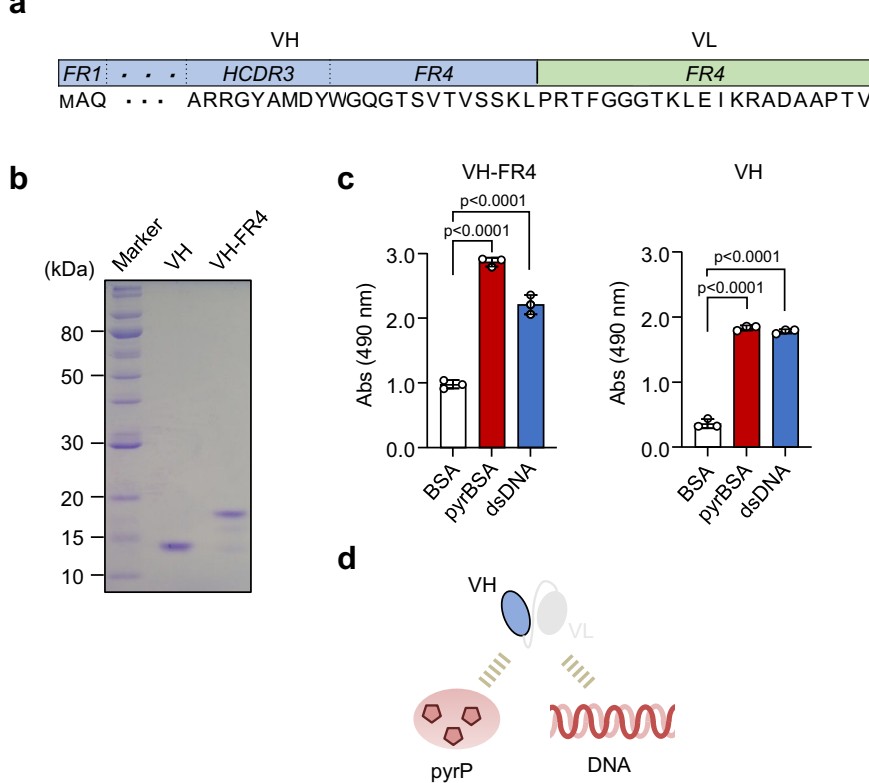

**Fig. 4 Discovery of a single VH domain that exhibits dual-specificity for both DNA and pyrP. a** Amino acid sequence of clone VH-FR4, lacking the peptide linker and most VL domains except for FR4. The sequences of FR1–FR3 of the heavy chain are abbreviated with dots. **b** SDS-PAGE analysis of two recombinant proteins, VH-FR4 and VH. They were expressed and purified from clone 5 as His$_6$-tagged recombinant proteins. M, molecular weight marker. **c** Binding of VH and VH-FR4 to antigens (BSA, pyrBSA, and DNA). Data are mean ± S.D. of triplicate samples (representative of three independent experiments). Dunnett's test (two-sided), relative to BSA. **d** Schematic representation of dual-specificity of VH for DNA and pyrP.

of the antibodies to the antigens (pyrBSA and dsDNA) is sensitive to the ionic strength. To further validate the contribution of arginine residues to the pyrrole/DNA cross-reactivity, arginine substitution mutants of VH DO1 and VH DO2 were generated by site-directed mutagenesis and tested for their binding to the antigens. For this purpose, the arginine residue in CDR2 was mutated to amino acids corresponding to germline sequences (VH DO2-R56N), and arginine residues in CDR3 were mutated to alanine (VH DO1-R97A, VH DO2-R96A, and VH DO2-R99A). The ELISA analysis showed that the mutation resulted in a dramatic reduction in the binding of the domain Abs to both pyrBSA and DNA (Fig. 6b, c). These results indicated that electrostatic interactions driven by the arginine residues in the VH domain Abs may be involved in the recognition of DNA/pyrP.

**Identification of pyrK-containing peptides recognized by VH domain Abs**. Next, we sought to identify the sequences within pyrBSA responsible for binding to the VH domain Abs. For this purpose, we used the VH DO2, which can be for both ELISA and immunoblotting (Supplementary Figs. 6). pyrBSA was digested with the Glu-C protease, which specifically hydrolyzes proteins on the carboxyl side of glutamate, and the digested peptides were separated and fractionated by reverse-phase high-performance liquid chromatography (HPLC). The ELISA analysis of the HPLC fractions showed that VH DO2 was immunoreactive with a fraction with a retention time of 45–50 min (Fig. 7a). The fraction was further separated into 10 fractions from which two pyrrolated peptides were identified by a matrix-assisted laser desorption ionization time-of-flight mass spectrometry (MALDI-TOF/MS) analysis: pyrBSA$_{300-320}$ (NLPPLTADFAEDK$^{pyr}$DVCKNYQE

(m/z = 2536. 35 $[M + H + H_2O]^+$) and pyrBSA$_{504-519}$ (K$^{pyr}$LF TFHADICTLPDTE (m/z = 1958.72$[M + H]^+$) (Supplementary Fig. 7, Fig. 7b, c). To gain an insight into the binding of the VH domain Abs to pyrrolated peptides, two recombinant peptides, BSA$_{300-320}$ and BSA$_{504-519}$, were expressed as maltose-binding protein (MBP)-fused peptides, followed by pyrrolation with butanedial and cleavage of MBP. Cysteine residues were replaced with serine residues to avoid peptide dimerization via disulfide bonds, yielding the pyrrolated recombinant peptides, pyrBSA$_{300-320}$ (GNLPPLTADFAEDK$^{pyr}$DVSK$^{pyr}$NYQE) and pyrBSA$_{504-519}$ (GK$^{pyr}$LFTFHADISTLPDTE) (Supplementary Fig. 8, Fig. 7d). A competitive ELISA with these peptides showed that VH DO2 binding to the pyrrolated proteins was dose-dependently inhibited by pyrBSA$_{300-320}$ but not by pyrBSA$_{504-519}$ (Fig. 7e). A further shortened peptide of pyrBSA$_{300-320}$, pyrBSA$_{310-320}$ (GEDK$^{pyr}$DVSK$^{pyr}$NYQE), showed a significant inhibitory effect on the binding of Abs to pyrP. A recombinant peptide (GQNK$^{pyr}$NVSK$^{pyr}$NYQQ), in which aspartate and glutamate in pyrBSA$_{310-320}$ (GEDK$^{pyr}$DVSK$^{pyr}$NYQE) were replaced with asparagine and glutamine, completely abolished the inhibitory effect (Supplementary Fig. 9, Fig. 7f).

On the other hand, the observations that the $N^\alpha$-acetyl derivative of pyrK (Supplementary Fig. 10a) and pyrrolated poly-L-lysine (Supplementary Fig. 10b) hardly inhibited the binding of VH DO2 to pyrP suggested that pyrK itself may not be the epitope structurally responsible for the recognition of pyrP by the VH domain Abs.

**X-ray crystallographic analysis of anti-DNA/pyrP scFvs**. Finally, to investigate the molecular details of the binding of the

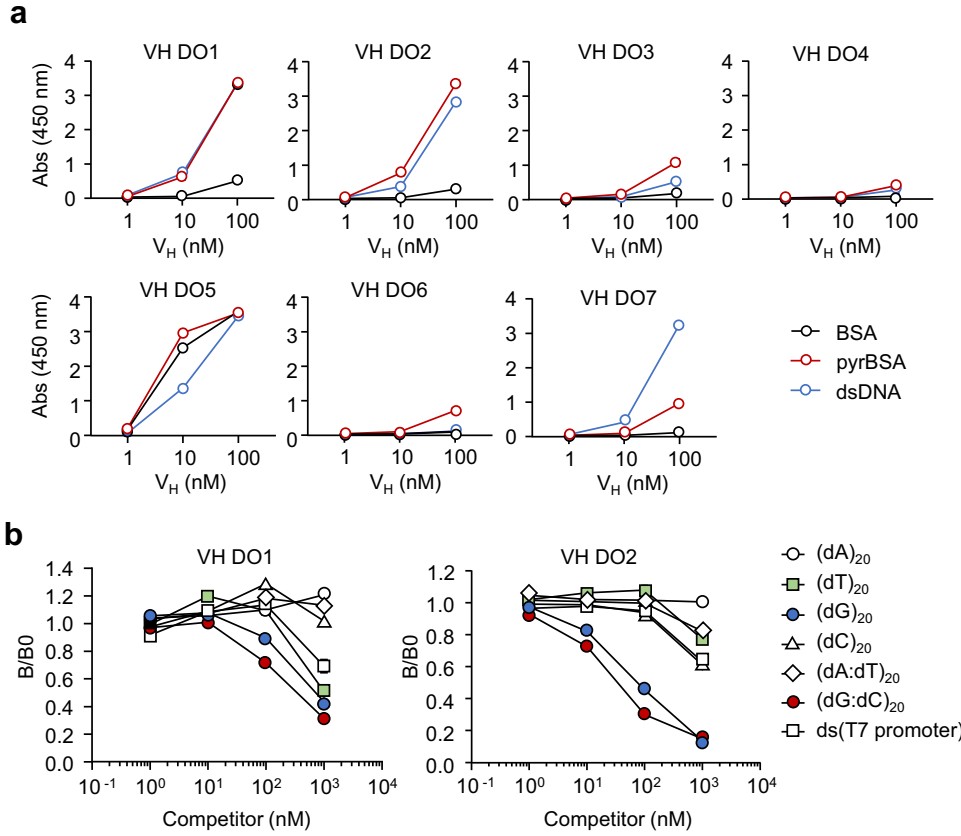

**Fig. 5 The VH domain primarily contributes to the recognition of DNA and pyrP. a** Binding of VH-His$_6$ to antigens (BSA, pyrBSA, and DNA). Antigens (1 μg/well) immobilized on an ELISA plate were incubated with VH-His$_6$ (1, 10, and 100 nM). Data are mean ± S.D. of triplicate samples (representative of three independent experiments). **b** Competitive ELISA for binding of VH DO1 (*left*) or VH DO2 (*right*) to DNA. The binding of VH-His$_6$ to DNA was evaluated in the presence of single-stranded DNA (ssDNA) or double-stranded DNA (dsDNA) (12.5, 25, 50, and 100 nM each). (dN)$_{20}$ indicates 20 bases of ssDNA, and (dN:dN)$_{20}$ indicates 20 base pairs of dsDNA. ds(T7 primer) is double-stranded DNA with sequence of 5′-TAATACGACTCACTATAGGG-3′. Data are mean ± S.D. of triplicate samples (representative of three independent experiments).

anti-DNA Abs to pyrP, we performed X-ray crystallography. We first attempted to perform crystallization experiments using the VH domain Abs DO1 and DO2; however, they were not crystallized. Hence, we adapted the "Fv-clasp" strategy[13] to crystallize the anti-DNA scFvs DO1 and DO2 as Fv-clasp and determined the crystal structure of the DO1-clasp (Fig. 8a, Table 1). An arginine residue (R97) in HCDR3, an important residue for cross-reactivity, was observed on the surface of the protein, giving the Ab a broad positive surface potential (Fig. 8b). The presence of such arginine residues was also confirmed in the antigen binding site of several anti-DNA Abs (Supplementary Fig. 11). To investigate the possible antigen-binding modes of the Abs, we performed the docking simulation using Autodock Vina with the DO1-clasp and ligands (pyrrolated peptides and DNA). For accurate docking simulations, the DK$^{pyr}$DVSK$^{pyr}$ peptide and (dG)$_3$ were used as ligands to reduce the number of molecular twists. The results showed that the pyrrolated peptide and DNA mainly interact with the VHs of the DO1-clasp (Supplementary Fig. 12a, b), especially with HCDR2, HCDR3 and FR around HCDR2 (Supplementary Figs. 12c, d). In FR, W47 and Y50 interacted with a pyrK residue in the pyrrolated peptide (Supplementary Fig. 12e) and guanine in DNA (Supplementary Fig. 12f). In addition, K58 interacted with an aspartic acid residue in the pyrrolated peptide and a phosphate group in DNA. Thus, the pyrK residue in the pyrrolated peptide corresponded to the guanine in DNA and the acidic amino acids to the phosphate groups.

**Discussion**

*N*-Pyrrolation of proteins is a charge-neutralizing reaction and is therefore accompanied by an increase in the net negative charges of the protein. In addition, proteins acquire an electrical conductivity via *N*-pyrrolation[9]. Due to these electrical properties, pyrP has been suggested to be recognized by anti-DNA Abs[8,14]. However, the mechanism underlying the binding of anti-DNA Abs to pyrP remains unknown. In the present study, to gain an insight into the molecular and structural basis of the dual specificity of the anti-DNA Abs, we used phage display to develop the recombinant Abs, including the scFvs and VH domain Abs. This technique enabled the discovery of the high-affinity, minimal Abs for DNA and pyrP, leading to the identification of the pyrK-containing epitope sequences in pyrP. Furthermore, we performed X-ray crystallography and computer-predicted simulations of the pyrK-containing peptide-scFv complexes to identify the key residues of Abs involved in the interaction with the antigens. Thus, this study partially resolved the classic problem of how anti-DNA Abs recognizes different antigens at the protein structure level.

In this study, based on the fact that SLE-prone MRL-*lpr* mice spontaneously develop anti-DNA titers[9], we generated a scFvs library from the mice and isolated several scFv clones specific to DNA. They were also shown to cross-react with pyrP. The sequence analysis of the anti-DNA/pyrP scFvs revealed that some of the scFvs (DO1, 3, 6–8) were composed of the V gene IGHV1-14*01 segment categorized in the J558 family. These results are

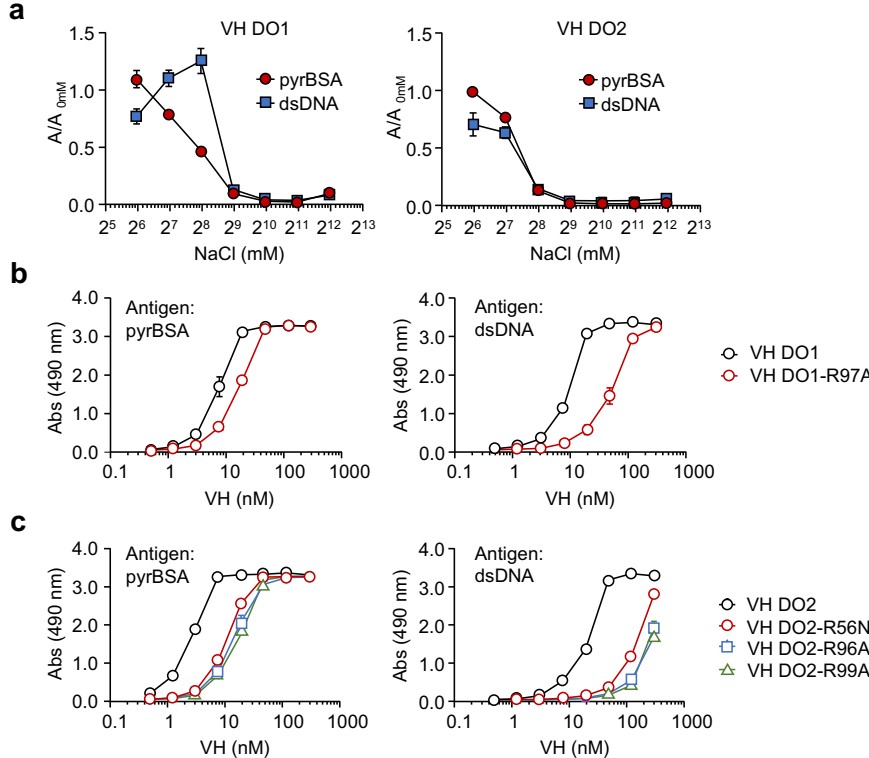

**Fig. 6 Significance of arginine residues in VH domain Abs for recognition of DNA and pyrP. a** Effect of ionic strength on antigen recognition of VH DO1 (*left panel*) and VH DO2 (*right*). Antigens (1 µg/well of pyrBSA or DNA) immobilized on ELISA plate were incubated with 10 nM VH-His$_6$ under the indicated concentrations of NaCl. **b** and **c** Binding of arginine-substituted mutants of VH DO1 (VH DO1-R97A) and VH DO2 (VH DO2-R56N, VH DO2-R96A, and VH DO2-R99A) to antigens (pyrBSA (*left panels*) and DNA (*right panels*)). Data are mean ± S.D. of triplicate samples (representative of three independent experiments).

consistent with the previous findings that the VH regions of the anti-DNA Abs are composed of specific V genes, such as the J558 gene family, in mice[15]. This consistency may be due to a common mechanism by which the development of the anti-DNA Abs in lupus-prone mice occurs by DNA and/or pyrP stimulating B cells to differentiate in a receptor-mediated response. On the other hand, our recent study showed a high IGHV2 usage and low IGHV1 usage in the anti-DNA/pyrP IgM repertoires[14]. Therefore, the VDJ gene rearrangements of the anti-DNA/pyrP IgG differ from those of the anti-DNA/pyrP IgM repertoire, suggesting that most of the VDJ gene rearrangement patterns in the anti-DNA/pyrP IgG repertoire may not be derived from the IgM repertoire.

In general, the anti-DNA Abs contains at least one critical arginine residue in HCDR3[4], which plays an important role in the recognition of DNA by Abs[5,12]. The HCDR3 sequence of the anti-DNA/pyrP scFvs indeed demonstrated that the scFvs contain multiple arginine residues (Fig. 2). These arginine residues in HCDR3 of the anti-DNA/pyrP scFvs are not found in the germline and may result from VDJ-recombination or somatic hypermutation. Previous studies have also shown that a monoclonal IgG specific for DNA/pyrP contains two arginine residues (arginine-arginine doublet) at the 5' end of HCDR3[9,16]. The presence of arginine in the anti-DNA Abs may be critical for the electrostatic interactions with pyrP[9], as *N*-pyrrolation confers an electronegativity and electronic properties to proteins. The importance of arginine residues in Abs has also been confirmed by X-ray crystallography and computer prediction simulations and discussed below (Fig. 8). On the other hand, it is also worth noting that the HCDR3 sequences of scFvs DO3–DO7 contain multiple tyrosine residues (Fig. 2). Tyrosine has been reported to

be involved in the recognition of antigens through hydrogen bonding and hydrophobic interactions that allow a favorable contact with the antigens[17–19]. Therefore, tyrosine residues in addition to the arginine residues in HCDR3, may mediate antigen recognition of the anti-DNA/pyrP Abs.

The sequence analysis of the anti-DNA/pyrP scFvs allowed us to discover a unique scFv composed of a VH domain and a light chain frame region (FR4) (Figs. 2, 4a). Based on this finding, we developed the anti-DNA VH domain Abs from a VH library in phage and found that the VH domain is sufficient for recognition of both DNA and pyrP. Competitive ELISA using nucleobase polymers also showed that the VH domain Abs are relatively specific for dG-containing polymers, such as (dG)$_{20}$ and (dG:dC)$_{20}$, compared to the others (Fig. 5b). These data suggest that the VH domain may be sufficient for the Abs specificity for both DNA and pyrP. Furthermore, we also observed that the binding of the VH domain Abs to pyrP and DNA was significantly inhibited by NaCl, and that the mutation of arginine residues to alanine or asparagine reduced the binding of the domain Abs to antigens (Fig. 6). These findings support the hypothesis that electrostatic interactions driven by the arginine residues in the VH domain Abs may be involved in recognition of DNA/pyrP.

To gain structural insights into the recognition of pyrP by the anti-DNA Abs, we sought to identify the pyrK-containing sequence of pyrBSA responsible for the binding to the anti-DNA/pyrP VH domain Abs. After proteolysis of pyrrolated BSA followed by HPLC separation, two pyrK-containing peptides, pyrBSA$_{300-320}$ and pyrBSA$_{504-519}$, were identified as candidate epitope sequences (Supplementary Figs. 7–9, Fig. 7b, c). These peptides contained pyrK at K312, K316, and K504. In our

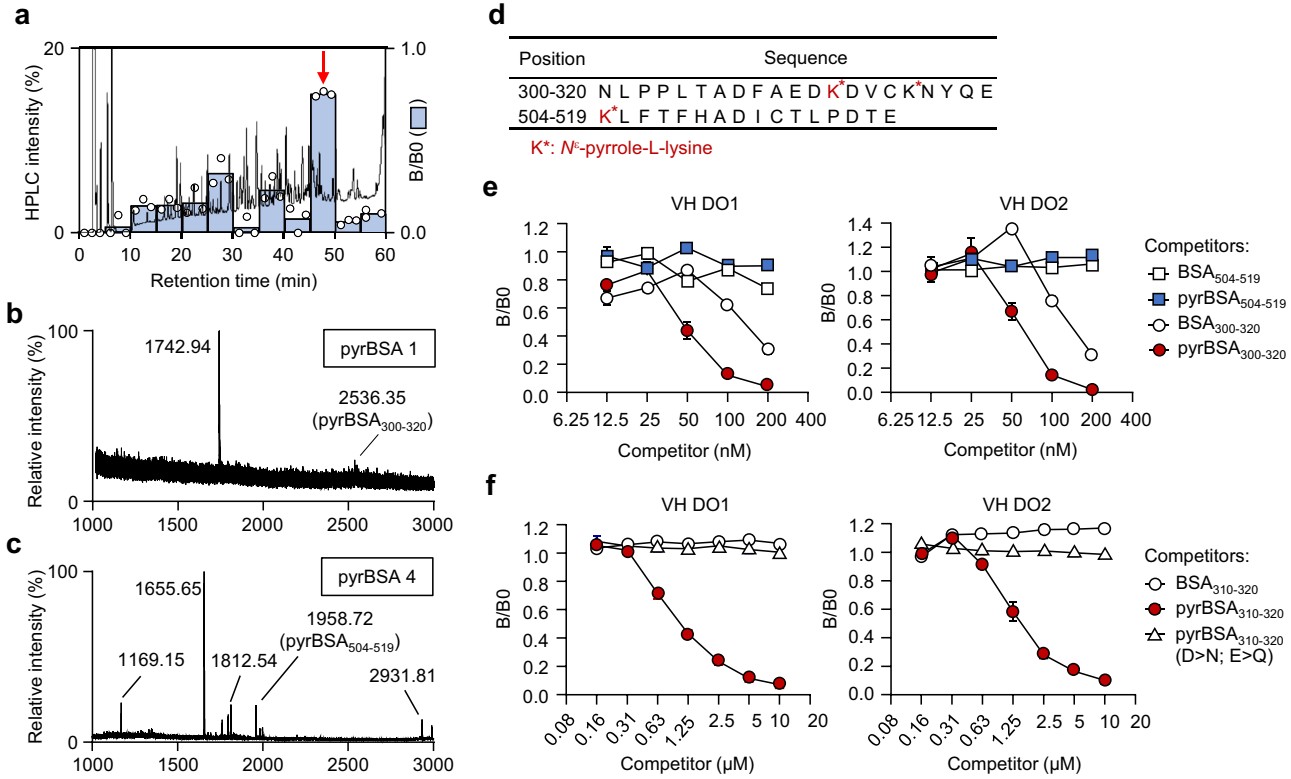

**Fig. 7 Identification of pyrK-containing peptides recognized by VH domain Abs. a** Fractionation of digested peptides from pyrBSA by HPLC and competitive ELISA. Bars indicate competition rates (B/B0) for competitive ELISA using VH DO2 as shown by mean of triplicate samples (representative of three independent experiments). Eluate was monitored by absorbance at 220 nm and collected every 5 min for 0–60 min. The fraction with retention times of 45–50 min is indicated by a red arrow. **b, c** MS spectra of fractions containing pyrK-containing peptides. The fraction with retention time of 45–50 min for pyrBSA-digested peptides was further separated into ten fractions and the MS spectra of fractions 1 and 4 (with retention time 45.0–45.5 and 46.5–47.0, respectively) are shown. **d** Amino acid sequence of identified pyrK-containing peptides. PyrK is shown as K* highlighted in red. **e, f** Competitive ELISA with native or pyrrolated recombinant peptides as competitors. Binding of VH DO1 (*left panel*) and VH DO2 (*right panel*) to pyrBSA was evaluated in the presence of $BSA_{504-519}$, $pyrBSA_{504-519}$, $BSA_{300-320}$, or $pyrBSA_{300-320}$ (**e**), and $BSA_{310-320}$, $pyrBSA_{310-320}$, and $pyrBSA_{310-320}$ (D > N; E > Q) (**f**). Data are mean ± S.D. of triplicate samples (representative of three independent experiments).

previous study, 25 lysine residues were identified as pyrrolation targets of pyrBSA[9]. However, the three lysine residues detected in this study are not among these 25 lysine residues. Although the exact reason for this discrepancy remains unknown, the two peptides identified in this study were relatively large molecules that may not have been detected in the previous study using nano-LC-ESI-Q-TOF MS.

Experiments with recombinant peptides further revealed that, instead of $pyrBSA_{504-519}$ containing one pyrK residue, only $pyrBSA_{300-320}$ containing two pyrK residues showed a specificity to the VH domain Abs (Fig. 7e). Furthermore, $pyrBSA_{310-320}$ (GEDK$^{pyr}$DVSK$^{pyr}$NYQE), which lacks the N-terminal decapeptide moiety of $pyrBSA_{300-320}$, still retained a binding activity to the VH domain Abs (Fig. 7f). Substitution of asparagine and glutamine for aspartate and glutamate in the peptide ($BSA_{310-320}$), respectively, completely abolished the binding activity of the domain Abs to pyrP (Fig. 7f). Therefore, $pyrBSA_{310-320}$ (GEDK$^{pyr}$DVSK$^{pyr}$NYQE) at this stage was identified as the minimal epitope sequence with specificity for the anti-DNA/pyrP VH domain Abs. These findings, along with the result that the VH domain Abs recognize neither Ac-pyrK nor poly-pyrK (Supplementary Figs. 10a, b), suggest that a sequence containing pyrK, rather than the pyrK residue alone, is involved in recognition of pyrP by Abs. Neutralization of the lysine residues by pyrrolation serves to alter the electric potential of the sequences, allowing them to be converted into a ligand for the anti-DNA/pyrP Abs. On the other hand, the modes of interaction between

the arginine residues in proteins and dG in DNA include (i) salt bridges between the arginine residues and phosphate groups and (ii) cation-π interactions between the arginine residues and nucleobases[20,21]. Therefore, we speculate that both pyrK and multiple acidic amino acids may be involved in the recognition by the anti-DNA Abs via cation-π interactions and salt bridge formation, respectively.

The X-ray crystallographic analysis (Fig. 8) revealed that the arginine residue in HCDR3 confers a strong positive surface potential on DO1. This feature is also observed with the other anti-DNA Abs A52 and DNA-1. Previous studies have shown that these anti-DNA Abs cross-react with heparan sulfate[22] and α-actinin[23]. Heparan sulfate contains negatively charged sulfate groups, and α-actinin is an acidic protein with a relatively low pI value (approximately 5). Therefore, the arginine residue of HCDR3 may play an important role in the multi-specificity of anti-DNA Abs against other antigens, such as pyrP. A docking simulation analysis indicated a similar binding mechanism of DO1-clasp with DNA and pyrP. Acidic amino acids in pyrrolated peptides or phosphate groups in DNA were predicted to interact with K58 on FRs around HCDR2. Other anti-DNA/pyrP Abs DO2 and DO4 do not contain K58 but instead contain R56 in HCDR2. These data suggest that basic amino acids around HCDR2, in addition to those in HCDR3, may contribute to the cross-reactivity with both DNA and pyrP.

In summary, to gain an insight into the specificity of the anti-DNA Abs for pyrP, we applied phage display technology that

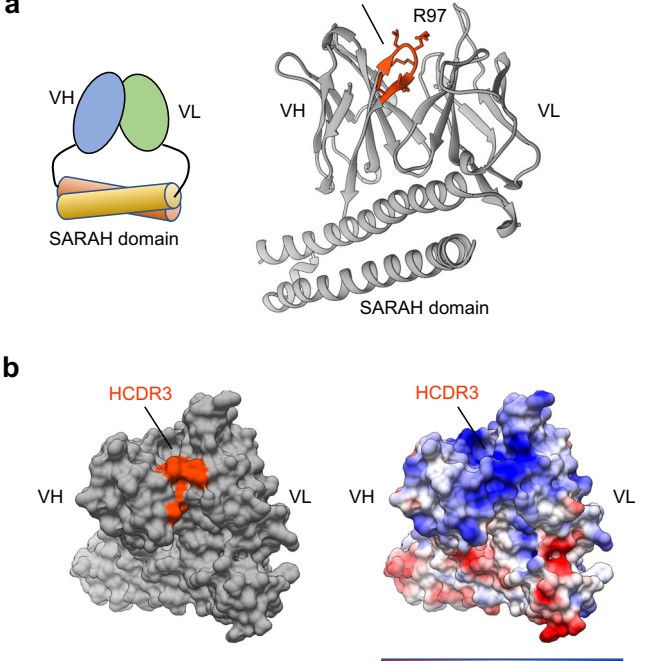

**Fig. 8 Crystal structure of DO1-clasp. a** Schematic representation of the Fv-clasp (*left*) and ribbon model of the DO1 clasp (*right*). Amino acids of HCDR3 are indicated and highlighted in orange. **b** Three-dimensional (3D) surface model of the DO1-clasp. HCDR3 is highlighted in orange (*left*), and the electrostatic surface potential is shown (*right*); red, negative; white, neutral; blue, positive.

## Table 1 Summary of X-ray data collection and refinement statics.

| Data collection | |
|---|---|
| Detector | EIGER16M |
| Wavelength (Å) | 0.899995 |
| Space group | $P2_12_12_1$ |
| Resolution (Å) | 47.182–1.990 (2.000–1.990)* |
| *Unit cell parameters* | |
| *a, b, c* (Å) | 71.66, 83.36, 112.58 |
| Total reflections | 312,822 (46,056) |
| Unique reflections | 92,507 (14,760) |
| Completeness (%) | 99.1 (97.9) |
| Redundancy | 3.38 (3.12) |
| $R_{meas}$ (%) | 15.0 (139.3) |
| CC 1/2 (%) | 0.993 (0.460) |
| <I/ σ (I)> | 5.89 (1.10) |
| *Refinement parameters* | |
| Resolution (Å) | 19.981–2.000 (2.071–2.000)* |
| No. of reflections | 48,192 (4714) |
| $R_{work}/R_{free}$ (%) | 0.2414/0.2981 |
| *No. of atoms* | |
| Proteins | 5187 |
| Water | 382 |
| *R.m.s. deviations* | |
| Bond length (Å) | 0.015 |
| Bond angles (°) | 2.32 |
| *Ramachandran plot* | |
| Favored (%) | 96.54 |
| Allowed (%) | 2.83 |
| Outliers (%) | 0.63 |

*The values in parentheses represent the data for the highest-resolution shell.

allows the development of the scFvs and VH domain Abs specific for both DNA and pyrP. Using these recombinant Abs, we identified the pyrK-containing peptide ligands that could selectively bind the Abs and suggested a plausible binding mechanism. The results of this study provide structural insights into the recognition of non-nucleic acid molecules by anti-DNA Abs and provide a platform for future engineering of anti-DNA Abs and the basis for therapeutic intervention of SLE.

## Experimental procedures

**Materials**. BSA, HSA, human transferrin, and human IgG were purchased from Fujifilm Wako (Osaka, Japan). Salmon sperm double-stranded DNA and polyK were purchased from Fujifilm Wako. 1,4-Butandial (BDA), 4-hydroxy-2-nonenal, 4-hydroxy-2-hexenal, and 4-oxo-2-nonenal were synthesized as previously reported[14]. Other aldehydes were purchased from the following sources: crotonaldehyde, 2-hexenal, and 2-decenal were purchased from Fujifilm Wako; acrolein was purchased from Tokyo Kasei; 2-pentenal, 2-heptenal, 2-octenal, 2-nonenal, and methylglyoxal were purchased from Sigma. All the other reagents used in the study were of analytical grade and obtained from commercial sources.

**Mice**. All experiments were performed according to the guidelines of the Animal Usage Committee of the Faculty of Agriculture, The University of Tokyo, and were approved by the committee. Twenty-one-week-old female MRL/lpr mice (MRL/MpJJmsSlc-*lpr/lpr*) were purchased from Japan SLC. The mice were housed in a temperature-controlled, pathogen-free room with light from 8:00 to 20:00 (daytime) and had free access to

standard food and water. We have complied with all relevant ethical regulations for animal use.

**In vitro modification of proteins and lysine derivatives**. The modification of proteins by aldehydes was performed by incubating proteins (1 mg/ml) with aldehydes (1 mM) in phosphate-buffered saline (PBS, pH 7.4) at 37 °C for 24 h. Any unreacted aldehydes were removed by dialysis against PBS. The pyrrolation of polyK was performed by incubating BDA (1 mM) with polyK (1 mg/ml) in PBS at 37 °C for 24 h. $N^\alpha$-acetyl-L-pyrrolelysine (Ac-pyrK) was prepared by incubating BDA and $N^\alpha$-acetyl-L-lysine (Ac-K) (5 mM each) in PBS at 37 °C for 6 h. The mixture was freeze-dried and extracted in methanol, followed by purification using reverse-phase HPLC.

**Construction of phagemid library for phage display**. Spleen cells were collected from a 21-week-old female MRL/lpr mouse. The total RNA of the spleen cells was extracted using the RNA Basic Kit (Fastgene), then cDNA was synthesized using the PrimeScript RT Master Mix (Takara-bio). The VH and VL genes of mouse immunoglobulin were amplified by PCR with VH and VL-specific primers under the following conditions: 94 °C for 2 min, followed by 30 cycles of 94 °C for 30 s, 55 °C for 30 s, and 72 °C for 1 min and 1 cycle of 72 °C for 2 min. The primers were designed based on a previous report[24] and modified for cloning into pSEX81(Supplementary Table 1). The PCR products were isolated by agarose gel electrophoresis and purified using the PureLink Quick Gel Extraction and PCR Purification Combo Kit (Thermo). The phagemid vector pSEX81 (Progen) and VL genes were digested with endonuclease *MluI* and *NotI* at 37 °C for 16 and 2.5 h, respectively. The digested VL genes were ligated into pSEX81 at 16 °C for 16 h with the Ligation high (Toyobo).

The VL-cloned pSEX81 was purified using the DNA clean & concentrator-5 (Zymo Research) and introduced into *E.coli* JM109 (Takara-bio) using ELEPO21 (Nepagene). The transformed *E.coli* was grown at 37 °C for 16 h on the 2YTAG (16 g/l tryptone, 10 g/l yeast extract, 5 g/l NaCl, 100 μg/ml ampicillin and 100 mM glucose) agar (1.5% (w/v)) plates. The phagemids were isolated using the PureLink Hipure Plasmid Midiprep Kit (Thermo-scientific). The VL-cloned pSEX81 and $V_H$ genes were digested with endonuclease *NcoI* and *HindIII* followed by ligation as already described. *E. coli* TG1 (Lucigen) was transformed with the scFv-cloned pSEX81 and grown at 37 °C for 16 h on the 2YTAG agar plates. The transformed bacteria were collected in a 2YTAG medium and stored as 15% (v/v) glycerol stock solution at −80 °C.

**Preparation of scFv-phage library**. *E. coli* transformed with pSEX81 was grown at 37 °C in 2YTAG medium until the $OD_{600}$ reached 0.4, infected with KM13 helper phage with a multiplicity of infection of 20 (MOI = 20) at 37 °C for 30 min, and further incubated with shaking for 30 min. *E. coli* was collected by centrifugation at $4000 \times g$ for 10 min and incubated in 2YTAK (16 g/l tryptone, 10 g/l yeast extract, 5 g/l NaCl, 100 μg/ml ampicillin, and 50 μg/ml kanamycin) medium at 30 °C with shaking for 16 h. The culture was centrifuged at $5000 \times g$ for 10 min, and the supernatant was mixed with PEG/NaCl solution (20% (w/v) polyethylene glycol 6000 and 2.5 M NaCl) at the ratio of 4:1 (v/v) and incubated on ice for 1 h. The scFv-phages were pelleted by centrifugation at $10,000 \times g$ for 30 min at 4 °C and resuspended in PBS. The scFv-phages were again precipitated using a PEG/NaCl solution and resuspended in PBS, and the bacterial debris was removed by centrifugation at $15,000 \times g$ for 5 min. To titer the phage library, *E. coli* XL1-blue MRF' (Agilent) at $OD_{600} = 0.4$ were infected with a serial dilution of the phage library and incubated overnight on a 2YTAG agar plate at 37 °C. The grown colonies were counted to calculate the phage titer.

**Biopanning with scFv-phage library**. DNA (100 μl of 10 μg/ml) in PBS was coated onto a well of a Maxisorp Immunoplate (Nunc) overnight at 4 °C. The well was washed three times with 300 μl of PBS containing 0.05% (v/v) Tween20 (PBST) and blocked with 370 μl of 2% (w/v) BSA in PBS for 2 h. After washing three times with PBST, 100 μl of the $1 \times 10^{12}$ phage forming unit (pfu) scFv-phage library, which was pre-incubated in a well blocked with BSA for 1 h, was added to the DNA-coated well and incubated for 2 h. The well was washed with PBST 10 (round 1) or 20 (round 2) times. Trypsin (200 μl of 10 μg/ml) in PBS was added to the well and incubated at 37 °C for 30 min to digest and detach the phages from the well. *E. coli* XL1-blue MRF' at $OD_{600} = 0.4$ in 5 ml of LB medium (BD Life Sciences) was infected with the collected phage at 37 °C for 30 min, then further incubated with shaking for 30 min. The infected bacteria were pelleted by centrifugation at $5000 \times g$ for 10 min and incubated overnight on 2YTAG agar at 37 °C. Preparation of the scFv-phage library and biopanning were repeated as already described.

**ELISA**. Antigens were coated overnight onto the wells of immunoplates at 4 °C. The plates were washed three times with 300 μl of PBST and blocked with 100 μl of 2% BSA in PBS for 1 h. After washing three times with PBST, 100 μl of the 1st antibody was added to the plates and incubated for 1 h. After washing three times with PBST, 100 μl of the 5000-fold diluted 2nd antibody labeled with HRP was added to the plates and incubated for 1 h. After washing three times with PBST, 100 μl of the HRP substrate was added to the plates. After incubation, 50 μl of 2 N $H_2SO_4$ was added to the wells to stop the color development, and the

absorbance at 450 nm for 3,3',5,5'-tetramethylenebendizine (TMB) or at 490 nm for *o*-phenylenediamine (PDA) was measured using a 2030 Multilabel Reader ARVO X3 (Perkin-Elmer). For ELISA with phage, $1 \times 10^9$ pfu phage libraries as the 1st antibody, anti-M13 major coat protein antibody (Cosmo bio) as the 2nd antibody, and TMB Ultra substrate (Thermo Scientific) as the HRP substrate were used. For ELISA with the Ab-fragments (scFv-His$_6$, VH-His$_6$, or VL-His$_6$), the Ab-fragment as the 1st antibody, anti-His-tag mAb-HRP-DirecT (MBL) as the 2nd antibody, and PDA (0.5 mg/ml) in 0.1 M citrate/phosphate buffer (pH 5.0) containing 0.003% $H_2O_2$ as the HRP substrate was used. For the competitive assay, the serial dilutions of the competitors were mixed with VH-His$_6$ and incubated for 30 min. For electrostatic analysis of VH, the VH was dissolved in 0.1 M phosphate buffer (pH 7.4) containing serially-diluted NaCl.

**Production of soluble scFv-G3P for screening**. *E. coli* XL1-blue MRF' was infected with phages in a trypsin solution at the second biopanning. The bacterial culture was diluted and inoculated onto 2YTAG agar to isolate a single bacterial colony. Ten colonies were randomly picked, resuspended in 2YTAG medium, and incubated overnight at 37 °C with shaking. The overnight culture (50 μl) was added to 1 ml of fresh 2YTAG medium and incubated with shaking at 37 °C for 2 h. The infected *E.coli* were collected by centrifugation at $4000 \times g$ for 10 min, and the expression of scFv-G3P was induced by incubating overnight in 2YT medium containing 50 μM isopropyl β-D-1-thiogalactopyranoside (IPTG, Wako) and 100 μg/ml ampicillin at 30 °C with shaking. The cultures were centrifuged at $4000 \times g$ for 10 min, and the supernatants were collected for the screening of scFv.

**Screening of scFv**. The wells of immunoplates were coated overnight at 4 °C with DNA (1 μg/ml). The plates were washed three times with 300 μl of PBST and blocked with 100 μl of 2% BSA for 1 h. After washing three times with PBST, 100 μl of the supernatants containing the soluble scFv-G3P was added to the plates and incubated for 1 h. After washing three times with PBST, 100 μl of 5000-fold-diluted anti-G3P antibody was added to the plates and incubated for 1 h. After washing three times with PBST, 100 μl of a 5000-fold-diluted anti-mouse IgG antibody labeled with HRP was added to the plates and incubated for 1 h. After washing three times with PBST, 100 μl of a TMB solution was added to the plates and incubated for 10 min. Fifty microlitres of 2 N $H_2SO_4$ was added to the plates to stop the color development, and the absorbance at 450 nm was measured.

**Sequence analysis of scFv genes**. Positive *E. coli* clones in the screening were grown in 2YTAG medium overnight at 37 °C. Phagemids were purified using the Plasmid Mini Kit (FastGene) and Sanger sequenced. The frame region (FR) and CDR of the antibody were determined by the Chothia numbering method[25] using Antibody Region-specific alignment software (AbRSA)[26].

**Cloning, expression, and purification of recombinant Ab-fragments, MBP-peptide, and TEV protease**. scFv, VH, and VL genes were subcloned from pSEX81 to pET22b(+) (Novagen) with a *N*-terminal PelB signal sequence and a *C*-terminal His$_6$-tag. The $V_H$ genes were cloned into the *NcoI* and *HindIII* sites, and the scFv and $V_L$ genes into the *NcoI* and *NotI* sites. *E.coli* BL21(DE3) (Agilent) transformed with the vectors was grown in LB medium containing 100 μg/ml ampicillin at 37 °C. After overnight culture, the cells were added into LB medium containing ampicillin in a flask and grown at 37 °C until the $OD_{600}$ reached 0.5. Expression was induced at 20 °C for 20 h by the addition of IPTG to a final concentration of 0.1 mM. The culture

was centrifuged at $4000 \times g$ for 10 min, and the supernatants were collected. The His$_6$-tagged Ab-fragment that leaked from the bacterial cells into the media was batch-bound at 4 °C for 1 h by rotating with Ni-NTA agarose resin (Fujifilm Wako) equilibrated with equilibration buffer (50 mM NaHPO$_4$ (pH 8.0), 300 mM NaCl, and 10 mM imidazole). The resin was washed with wash buffer (50 mM NaHPO$_4$ (pH 8.0), 300 mM NaCl, and 20 mM imidazole), and the Ab-fragment was eluted with elution buffer (50 mM NaHPO$_4$ (pH 8.0), 300 mM NaCl, and 250 mM imidazole). The buffer was exchanged into PBS using a PD MiniTrap G-25 column (Cytiva).

The genes of MBP with the *C*-term TEV protease cleavage site were cloned into pET28a carrying the *N*-term His$_6$-tag followed by the cloning of the peptide genes into the vectors. *E. coli* BL21(DE3) transformed with the vectors was grown in an LB medium containing 50 µg/ml kanamycin at 37 °C. After overnight culture, the cells were added into LB medium containing kanamycin in a flask and grown at 37 °C until the OD$_{600}$ reached 0.8. Expression was induced at 37 °C for 3 h by the addition of IPTG to a final concentration of 0.4 mM. The culture was centrifuged at $4000 \times g$ for 10 min, and the cells were collected, resuspended in equilibration buffer, and disrupted by sonication for 5 min. The cell lysate was centrifuged at 4 °C, $12,000 \times g$ for 30 min, and the supernatant was collected. The supernatant was poured into a column containing Ni-NTA agarose resin equilibrated with an equilibration buffer. After washing the resin with wash buffer, the MBP-peptide was eluted with elution buffer.

The expression and Ni-NTA purification procedure of TEV protease were the same as those of the MBP-peptide, except that the expression was induced in LB medium containing 100 µg/ml ampicillin for 3 h by the addition of IPTG to a final concentration of 0.1 mM, and that the elution buffer contained 1 mM dithiothreitol (DTT). Ammonium sulfate was added to the eluted TEV protease to a final concentration of 40% (w/v) and precipitated overnight at 4 °C. The precipitants were pelleted by centrifugation at 4 °C, $12,000 \times g$ for 30 min, and dissolved in 50 mM Tris-HCl (pH 8.0). The solution was further purified by size-exclusion chromatography using AKTA pure 25 (Cytiva) equipped with a Superose Increase 6 10/300 GL column (Cytiva) equilibrated with 50 mM Tris-HCl (pH 8.0).

**Site-directed mutagenesis of VH domain Abs and cloning of MBP-peptide.** Site-directed mutagenesis of the VH domain Abs and cloning of the MBP-peptide were performed based on the quick-change system. The pET22b(+) vectors carrying the VH genes or the pET28a vector carrying the MBP gene were amplified by PCR with the primers containing the mutation sequences or the peptide sequences. The amplified vectors were assembled using the NEBuilder Hifi DNA assembly Master Mix (NEB), and *E. coli* was transformed with the vectors.

**Protein digestion.** BSA and pyrBSA (25 mg/ml each) were denatured at 37 °C for 1 h in a buffer containing 50 mM ammonium bicarbonate, 8 M urea, and 0.1 M DTT. After cooling to room temperature, iodoacetamide (0.3 M) was added to the solutions and incubated at room temperature in the dark for 30 min. The solutions were 20-fold diluted with 50 mM ammonium bicarbonate buffer, and Glu-C protease (Promega) was added to the solution at the protease:protein ratio of 1:50 (w/w). The solutions were incubated overnight at 37 °C. The digested peptides were fractionated using reverse-phase HPLC. The collected fractions were dried using a centrifugal evaporator and dissolved in water.

**HPLC.** The purification of Ac-pyrK and the peptides was performed by reverse-phase HPLC equipped with a Sunniest RP-AQUA column (5 µm, $20 \times 250$ mm, ChromaNik Technologies). Ac-pyrK was eluted with a gradient of water containing 0.1% (v/v) trifluoroacetic acid (TFA) (solvent A) and acetonitrile containing 0.1% TFA (solvent B) at the flow rate of 10 ml/min (0–15 min, 30–90% B). The protein-digested peptides were eluted with solvent A and water containing 80% (v/v) acetonitrile and 0.1% (v/v) TFA (solvent C) at the flow rate of 1.0 ml/min (0–50 min, 5–55% C; 50–53.5 min, 55–95% C, 53.5–60 min, 95% C). The eluates were collected at 5-min or 30-s intervals. The pyrrolated peptides were eluted with solvent A and solvent B at the flow rate of 4.0 ml/min (0–15 min, 5–90% B). All the analyses were monitored at an absorbance of 220 nm.

**Mass spectrometry analysis.** The peptide-containing solutions were mixed with a matrix solution containing matrix compounds, 30% (v/v) acetonitrile, and 0.1% (v/v) TFA at the ratio of 1:1. Saturated α-cyano-4-hydroxycinnamic acid was used as the matrix for the digested peptides and 20 mg/ml 2,5-dihydroxybenzoic acid for the recombinant pyrrolated peptides. A 0.5 µl aliquot of the mixture was spotted on a MALDI ground steel plate (Bruker Daltonics) and dried. A MALDI-TOF/TOF mass analysis was performed using the Autoflex Speed TOF/TOF system (Bruker Daltonics) in the reflector positive or LIFT mode. A data analysis was performed using FlexAnalysis 3.4 software (Bruker Daltonics).

**Preparation of pyrrolated peptides.** The purified MBP-peptides were incubated overnight with BDA (2.5 eq of the amino groups on MBP-peptide) at 37 °C and dialyzed against 10 mM Tris-HCl (pH 8.0). The pyrrolated MBP-peptides were overnight incubated with TEV protease (2% (w/w; MBP-peptide/protease)) and DTT (1 mM) at room temperature. The cleaved His$_6$-tagged MBP and TEV protease were removed with Ni-NTA resin. The peptides were purified by reverse-phase HPLC and analyzed by MALDI-TOF/TOF mass.

**Preparation of Fv-clasps.** The genes of the SARAH domains for the VH or VL constructs were synthesized by two-step overlap-extension PCR. Two sets of four primers (SARAH_1, 2, 3_VH, and 4 or SARAH1, 2, 3_VL, and 4 (Supplementary Table 2) were mixed, respectively, in an equimolar ratio. The 1st PCR was carried out in a solution containing 1× KOD One PCR Master Mix (Toyobo) and a set of four primers (2 nM each) under the following conditions: 5 cycles of 98 °C for 10 s, 55 °C for 5 s, and 68 °C for 1 s. The genes were amplified by the 2nd PCR with SARAH_1 and SARAH_4 primers at a final concentration of 200 nM under the following conditions: 35 cycles of 98 °C for 10 s, 55 °C for 5 s, and 68 °C for 1 s. The genes of V$_L$ and the SARAH domain for the VL were cloned into pET28a and VH, and the SARAH domain for the VH was cloned into pET28a carrying the *N*-term His$_6$-tag and TEV protease cleavage site. The production and refolding of the Fv-clasps were conducted as previously described[13]. After refolding, the Ni-NTA purification and His$_6$-tag removal procedures were performed in the same manner as those of the MBP-peptide, except that the buffers did not contain DTT and the His$_6$-tag removal was conducted in a buffer containing 50 mM Tris-HCl (pH 8.0) at 4 °C. The Fv-clasp was further purified by SEC using a Superose Increase 6 10/300 GL column equilibrated with 50 mM Tris-HCl (pH 8.0) and 150 mM NaCl and concentrated using Vivaspin 6 (MWCO 10,000 Da, Sartorius).

**Crystallization, data collection, and structure determination of DO1-clasp**. Crystallization experiments were conducted using the sitting-drop vapor-diffusion method at 20 °C. One µl of the DO1-clasp solution (3 mg/ml) was mixed with 1 µl of precipitant solution containing 0.1 M Tris-HCl (pH 7.5), 25% (v/v) PEG 3350, and 0.2 M LiNO$_3$. The mixture was then equilibrated against 500 µl of the precipitant solution in the reservoir at 20 °C for 1 week. The crystals were flash-cooled, and their X-ray diffraction data were collected at BL44XU of SPring-8 (Hyogo, Japan). The X-ray diffraction data were indexed, integrated, and scaled using the XDS program[27]. The initial structural models of DO1 were obtained by molecular replacement with Molrep[28] using the coordinates predicted by alphafold2[29] as the template. Further model building and refinement were performed with REFMAC5[30], Coot[31], and Phenix[32]. The molecular structures were visualized using UCSF Chimera[33].

**Docking simulation**. A docking simulation analysis was performed using the AutoDock Vina program[34]. To reduce the number of torsions in the structure of the molecules under 32, the DK$^{pyr}$DVSK$^{pyr}$ peptide and (dG)$_3$ were used as ligands.

**Statistics and reproducibility**. All results were confirmed with three independent experiments, with the data of one representative experiment being presented. Data are represented as mean with standard deviation (S.D.) of triplicate samples as described in the respective figure legends. Statistical analyses were performed using Prism Software v6.07 (GraphPad), with a $p$-value of <0.05 being considered significant. Differences between the two groups were analyzed by two-tailed Student's $t$-test. Multiple comparisons between groups were made using Tukey–Kramer tests or Dunnett's test. The number of replicates was determined based on previous studies.

**Reporting summary**. Further information on research design is available in the Nature Portfolio Reporting Summary linked to this article.

## Data availability
Uncropped blot/gel images are included in Supplementary Fig. 13. The X-ray structure of the DO1 Fv-clasp fragment was deposited in the PDB, accession number 8HYL. The source data behind the graphs in the paper can be found in Supplementary Data 1, and any remaining information can be obtained from the corresponding author upon reasonable request.

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

## Acknowledgements
The X-ray diffraction data were collected at the Osaka University beamline BL44XU of SPring-8 (Harima, Japan) (Proposal No. 2022A6721 and 2022B6721) with the kind assistance of the beamline scientists. This study was supported by JSPS KAKENHI Grant Numbers JP17H06170, JP26111011, JP20K15467, and by AMED under Grant Number

JP19gm0910013h0003. This research was partially supported by the Platform Project for Supporting Drug Discovery and Life Science Research (Basis for Supporting Innovative Drug Discovery and Life Science Research (BINDS)) from AMED under Grant Number JP19am0101076 (support number 1496).

## Author contributions

Y.A., M.I., T.S., K.Y., P.L., and K.N. performed the experiments. P.L. and K.N. performed the X-ray data collection and structure determination of DO1-clasp. Y.A., J.D., and H.U. designed the experiment of phage display. Y.A., M.I., K.N., and K.U. wrote the manuscript. K.U. contributed to the funding acquisition. K.U. supervised and coordinated the research.

## Competing interests

The authors declare no competing interest.

## Inclusion and ethics statement

The research presented here adhered to the ethical and inclusivity standards consistent with the corresponding author's institutional and internal review board policies.
