## [Peer Review File · Communications Biology]

Reviewers' comments:

Reviewer #2 (Remarks to the Author):

The paper by Anan et al presents a study on the cross-reactivity of anti-DNA murine antibodies for pyrrolated proteins. The manuscript is well written. It is interesting that VH alone is sufficient for binding to pyrP. There is a crystal structure of an antibody fragment with a pyrK peptide that provides structural insights into the binding of this particular antibody. Overall, this paper should be interesting for people studying the specificity of antibodies in the context of autoimmune diseases or antibody drug development.

- In general, it is well known that excessive positive charge in the CDRs is related to non-specific binding. To what extent is the binding to both DNA and pyrP polyspecificity (binding of antibodies to a subset of defined targets) or polyreactivity (non-specific sticking to various biomolecules)?
- The authors used mostly ELISA to measure the antibody binding. Is there also kinetic data (BLI, SPR) about the binding of the antibodies to pyrP versus DNA? Are there differences in the on and off binding constants?
- Fig 5 – It seems that the binding is very sensitive to the ionic strength based on experiments with NaCl only. What is the impact serum on the antibody binding to the pyrBSA and dsDNA?
- Fig 5 – The mutation of arginine to asparagine or alanine reduces the binding. What will be the impact of a mutation arginine to lysine or arginine to aspartic acid?
- How do the VH CDR murine sequences compare to human CDR sequences? Are there consensus CDR sequence motifs (except for the arginine residues) that are common for antibodies that share specificity for DNA and pyrP?
- If the authors agree, the summary of the X-ray data in Table 2 could be moved to the supplementary data.

Reviewer #3 (Remarks to the Author):

The genesis of the submission is based on the structural insights of anti-DNA antibodies derived from SLE prone mice against pyrP. Phage display panning was carried out to identify the monoclonal scfv clone. Structural analysis using crystal structures was also carried out. The authors denoted the independent nature of the VH domain to be mainly responsible for the interaction.

Comments:

I would like to seek some clarification on the clone selection. The authors wrote "After two rounds of biopanning, 10 clones of the scFv phage were randomly selected" and further wrote "Of the 18 clones, 8 clones consisted of full-length variable heavy (VH) and variable light (VL) regions, and 3 clones contained internal amber stop codons". I realise the author wrote that the selection was repeated. So the 18 clones is from the second round or its 10 +8 clones? Also from the 18, 8 were full and 3 had stops. What about the remaining 7 clones? I am assuming they were empty clones. If yes, please state to avoid confusion.

However looking at Fig1, there are 20 clones that were tested. This does not tally with the previous statement of 10 and 18.

There is no actual need to state the subgroups of the antibody sequences (J558 and 7183). It would be better to state the D and J gene segment identifiers.

The authors mentioned 8 clones were identified where 5 were from J558 and 1 was from 7183. What about the remaining 2?

Fig 2 does not actually show affinity but more of binding function. There is no on and off rate calculated. Therefore the use of the term affinity in this context is not accurate.

There is no error bar for Fig 2.

The statement "VH with a light chain frame region lacking the peptide linker and most of the VL region (VH-FR4)" is confusing. I think it would be better to rephrase is as a truncated scFv with only the VH domain present.

"VH with a frame region of a light chain (VH-FR4)". What is the justification of adding the FR4 of the LC to the VH domain clone?

From the ELISA, it appears the addition of the LC FR4 to the VH clone increased the BSA binding. Was this explained?

The authors split the VH and VL domains for testing and concluded the VH domain to be predominantly responsible. However, from the ELISA test only 3 of the 8 VH clones showed binding with clone DO7 only binding dsDNA with no cross reaction to pyrBSA whereas the remaining 4 did not show significant binding. As the clones are all unique in terms of the CDRs to provide a generalized statement is rather misleading.

Additionally, there should be a comparison between the VH only clones with the full scfv as the general perception is that a synergistic effect of the VH and VL domains are responsible for improved binding. I also acknowledge in some cases the VH domain alone is sufficient but the lack of comparison between the scFv and VH only makes the conclusion weak.

The conclusion of the VH domains being predominant in the binding can only be mentioned for 2 clones which are DO1 and DO2. I think it would be best to refocus the claims to mention clearly only this two clones as it would not be really comparable with the other clones. Additionally the phenomena of the VH being the dominant domain is only isolated to this two clones only.

The alanine mutation experiment was greatly appreciated to prove the role of the specific amino acids in binding.

Dear Reviewers:

We thank the reviewers for their constructive comments and suggestions on the manuscript. They were very helpful to provide further evidence to support our claim and to emphasize the physiological and pathophysiological relevance of our findings. We carefully revised the manuscript according to the instructions by the editor and the reviewers. We sincerely hope that our responses and revisions allow this paper to achieve a priority sufficient for publication in *Communication Biology*.

Below is a detailed point-by-point response to all comments by the reviewers (reviewer's comment in blue and our response in black).

Response to Reviewer #2:

The paper by Anan et al presents a study on the cross-reactivity of anti-DNA murine antibodies for pyrroled proteins. The manuscript is well written. It is interesting that VH alone is sufficient for binding to pyrP. There is a crystal structure of an antibody fragment with a pyrK peptide that provides structural insights into the binding of this particular antibody. Overall, this paper should be interesting for people studying the specificity of antibodies in the context of autoimmune diseases or antibody drug development.

Response: We are very pleased to receive favorable reviews from the reviewer.

- In general, it is well known that excessive positive charge in the CDRs is related to non-specific binding. To what extent is the binding to both DNA and pyrP polyspecificity (binding of antibodies to a subset of defined targets) or polyreactivity (non-specific sticking to various biomolecules)?

Response: As shown in the Extended Data Figs. 1 and 2, all seven anti-DNA scFVs are highly specific to pyrroled proteins. On the other hand, in our previous paper (Ref. 9), we showed that lysine pyrrolation is associated with an increase in the net negative charges of proteins. It has been suggested that this may give rise to the structural similarity to the dsDNA backbone, which is composed of negatively charged phosphate groups. Thus, anti-pyrP/DNA antibodies are specific to pyrroled proteins but could be polyreactive to molecules that have similar electrostatic potentials.

- The authors used mostly ELISA

Supplementary Data for Review: Figure 1. Surface plasmon resonance measurements. The interactions of BSA or pyrroled BSA with VH Abs (DO1 and DO2) immobilized by amine coupling in the lanes of a BIAcore sensor chip NTA were monitored by the single-cycle kinetics method. The surface plasmon resonance assays were performed using a BIAcore T100 instrument (GE Healthcare). The VH His-tag were immobilized on a sensor chip NTA (GE Healthcare) at a density of 1000 response units (RU). The interaction between the immobilized VH Abs and pyrroled BSA (6.25~100 ng/ml) was examined at 25°C with a flow rate of 30 μ l/min by a single-cycle kinetics analysis program. HBS-P (10 mM HEPES-NaOH (pH 7.4), 150 nM NaCl, 0.05% Tween 20) was used as the running buffer. The response curves obtained from injecting buffer only and from the control flow cell (without immobilized VH Abs) were subtracted from the VH Abs-immobilized cell to correct for any nonspecific binding. BIAevaluation software (version 4.1) was used to perform the kinetic analysis.

to measure the antibody binding. Is there also kinetic data (BLI, SPR) about the binding of the antibodies to pyrP versus DNA? Are there differences in the on and off binding constants?

Response: This was also pointed out by another reviewer, suggesting that using the term “binding” is more accurate in this context. We agree with the reviewer’s comment and have changed it from “affinity” to “binding” in the revised manuscript. We have also tried to measure the affinity of antibody to pyrP and DNA using SPR. We have so far obtained data on VH Abs (DO1 and DO2) binding to pyrP (**Supplementary data for review: Fig. 1**), showing the binding of pyrP to the VH Abs. However, measurement of the binding to DNA has not been completed because of the long-term unavailability of sensor chip for DNA. We will continue this experiment in the future, so we would like to report on kinetic data, including the data on VH binding to pyrP, on another occasion.

- Fig 5 – It seems that the binding is very sensitive to the ionic strength based on experiments with NaCl only. What is the impact serum on the antibody binding to the pyrBSA and dsDNA?

Response: Based on the reviewer’s comment, we carried out the experiment regarding the effect of serum addition on the antibody binding to the ligands (pyrBSA and dsDNA). The data showed that the serum inhibited the antibody binding to the pyrBSA and dsDNA in a dose-dependent manner (**Supplementary data for review: Fig. 2**). The data also support our hypothesis that the binding of the antibodies to the antigens (pyrBSA and dsDNA) is sensitive to the ionic strength.

Supplementary Data for Review: Figure 2. Effect of serum on binding of VH DO1 and DO2 Abs to pyrBSA or dsDNA. pyrBSA or dsDNA (10 µg/ml) immobilized on ELISA plate was incubated with VH Abs (10 nM) in the presence of mouse serum (0.016, 0.08, 0.4, 2, or 10%).

- Fig 5 – The mutation of arginine to asparagine or alanine reduces the binding. What will be the impact of a mutation arginine to lysine or arginine to aspartic acid?

Response: We appreciate the reviewer’s constructive comments. This experiment is to show the involvement of arginine in the antibody binding to the antigens and we believe that, at least in the current study, the mutation of arginine to asparagine or alanine may be enough. However, for further detailed binding mechanism, a mutation of arginine to lysine or arginine to aspartic acid should also be considered.

- How do the VH CDR murine sequences compare to human CDR sequences? Are there consensus CDR sequence motifs (except for the arginine residues) that are common for antibodies that share specificity for DNA and pyrP?

Response:

As far as we know, CDRs of both murine and human anti-DNA Abs don’t contain any consensus sequential motif except for the arginine residues.

- If the authors agree, the summary of the X-ray data in Table 2 could be moved to the supplementary data.

Response: We agree to move the summary of the X-ray data in Table 2 to the supplementary

data (Extended Data Table 3).

Response to Reviewer #3:

The genesis of the submission is based on the structural insights of anti-DNA antibodies derived from SLE prone mice against pyrP. Phage display panning was carried out to identify the monoclonal scFv clone. Structural analysis using crystal structures was also carried out. The authors denoted the independent nature of the VH domain to be mainly responsible for the interaction.

Comments:

I would like to seek some clarification on the clone selection. The authors wrote “After two rounds of biopanning, 10 clones of the scFv phage were randomly selected” and further wrote “Of the 18 clones, 8 clones consisted of full-length variable heavy (VH) and variable light (VL) regions, and 3 clones contained internal amber stop codons”. I realise the author wrote that the selection was repeated. So the 18 clones is from the second round or its 10 +8 clones? Also from the 18, 8 were full and 3 had stops. What about the remaining 7 clones? I am assuming they were empty clones. If yes, please state to avoid confusion.

Response: This was due to an error in counting the numbers of clones. The sentence should have been as follow: “Of the 12 clones, 8 clones (clones 3, 4, 6, 9, 13, 14, 17, and 18) consisted of full-length variable heavy (VH) and variable light (VL) regions, and 3 clones (clones 8, 12, and 16) contained internal amber stop codons”. This sentence and the sentence “One remaining clone (clone 5) is described later.” have been added in the revised manuscript (page 5, first paragraph).

However looking at Fig1, there are 20 clones that were tested. This does not tally with the previous statement of 10 and 18.

Response: The sentences in the original manuscript were mistakenly described. We rewrote them in the revised manuscript as follows: “After two rounds of biopanning, 10 clones of scFv phage were randomly selected from the first screening and the other 10 clones of the scFv phage were randomly selected from the second screening, and the binding of these 20 clones to DNA was evaluated (**Fig. 1d**). They are followed by the sentence “Of the 12 clones, 8 clones.....” above, which has been corrected in response to the reviewer’s remark. These sentences accurately describe the experiments that were performed.

There is no actual need to state the subgroups of the antibody sequences (J558 and 7183). It would be better to state the D and J gene segment identifiers.

Response:

Although the reviewer commented that there is no actual need to state the subgroups of the antibody sequences (J558 and 7183), we prefer to retain the use of the subgroups of the antibody sequences to emphasize that the scFvs selected in this study consisted of the specific V genes characteristic to anti-DNA Abs.

The authors mentioned 8 clones were identified where 5 were from J558 and 1 was from 7183. What about the remaining 2?

Response:

The remaining two clones (DO4 and DO5) proved not to be classified into any gene family.

Fig 2 does not actually show affinity but more of binding function. There is no on and off rate calculated. Therefore the use of the term affinity in this context is not accurate.

Response: We agree with the reviewer's comment and have changed it from "affinity" to "binding" in the revised manuscript.

There is no error bar for Fig 2.

Response: It has error bars. However, they are so small that they are covered with symbols.

The statement "VH with a light chain frame region lacking the peptide linker and most of the VL region (VH-FR4)" is confusing. I think it would be better to rephrase is as a truncated scFv with only the VH domain present.

Response: We agree with the reviewer's comment and rephrased it following the reviewer's suggestion (page 6, line 1).

"VH with a frame region of a light chain (VH-FR4)". What is the justification of adding the FR4 of the LC to the VH domain clone?

Response: In VH-FR4, VH and VLFR4 are linked at the restriction site of HindIII (AAGCTT). Therefore, it can be speculated that this clone was artificially generated during phage display experiment.

From the ELISA, it appears the addition of the LC FR4 to the VH clone increased the BSA binding. Was this explained?

Response: As pointed out by the reviewer, although these experiments with VH-FR4 (left panel) and VH (right panel) were performed independently, VH-FR4 was slightly more specific to BSA than VH. This means that FR in VH-FR4 affected the antigen binding, but the reason remains unclear. We think that this speculation may not be necessary in the manuscript. However, if the reviewer/editor feels otherwise, we will abide by their decision.

The authors split the VH and VL domains for testing and concluded the VH domain to be predominantly responsible. However, from the ELISA test only 3 of the 8 VH clones showed binding with clone DO7 only binding dsDNA with no cross reaction to pyrBSA whereas the remaining 4 did not show significant binding. As the clones are all unique in terms of the CDRs to provide a generalized statement is rather misleading.

Response: We agree with the reviewer's comment that a generalized statement of the VH domain to be predominantly responsible is rather misleading. Therefore, the sentences on Fig. 4a were rephrased in the revised manuscript (page 6, lines 12-16).

Additionally, there should be a comparison between the VH only clones with the full scfv as the general perception is that a synergistic effect of the VH and VL domains are responsible for improved binding. I also acknowledge in some cases the VH domain alone is sufficient but the lack of comparison between the scFv and VH only makes the conclusion weak.

Response:

The ELISA experiments on scFv (Fig. 2b), VH (Fig. 4a), and VL (Extended Data Fig. 3) were performed simultaneously. Therefore, the results of these experiments were comparable.

The conclusion of the VH domains being predominant in the binding can only be mentioned

for 2 clones which are DO1 and DO2. I think it would be best to refocus the claims to mention clearly only these two clones as it would not be really comparable with the other clones. Additionally the phenomena of the VH being the dominant domain is only isolated to these two clones only.

Response:

We agree with the reviewer's comment that the phenomena of the VH being the dominant domain is only isolated to these two clones only. Hence, the conclusion of the VH domains being predominant in the binding was rephrased to "Thus, we established that some VH domains, at least VH DO1 and DO2, could recognize the antigens (DNA/pyrP) without VL domains." in the revised manuscript (page 6, lines 18-20).

The alanine mutation experiment was greatly appreciated to prove the role of the specific amino acids in binding.

Response: Because the effects of amino acid substitutions were clearly shown in Fig. 4, we may not need further amino acid substitution study, at least for this manuscript. However, as suggested by the reviewer, the alanine mutation experiment may further prove the role of the specific amino acids in the binding. We would like to take this proposal into account in the project on protein N-pyrrolation.

REVIEWERS' COMMENTS:

Reviewer #2 (Remarks to the Author):

The authors have addressed all my concerns.

I have only two remaining minor points:

- the table with the summary of X-ray data appears twice in the revised manuscript (as Table 2 and Ext Table 3)
- it would be good if the supplementary figures for review included in the rebuttal are actually included as extended data published with the manuscript

Reviewer #3 (Remarks to the Author):

I greatly appreciate the responses by the authors to the comments raised. All my concerns have been addressed.

Point-by-point reply: Anan et al. Manuscript ID: COMMSBIO-23-1271-T

Dear Reviewers:

We thank the reviewers for their constructive comments on the manuscript. Following the suggestion by Reviewer #2, we have removed Table 2 from the revised manuscript and included Supplementary Figures for Review as Extended Data in the revised manuscript. We sincerely hope that our responses and revisions allow this paper to achieve a priority sufficient for publication in *Communications Biology*.

Below is a detailed point-by-point response to all comments by the reviewers (reviewer's comment in blue and our response in black).

Response to Reviewer #2:

The authors have addressed all my concerns. I have only two remaining minor points:

Response: We sincerely thank the reviewer for a positive evaluation.

- the table with the summary of X-ray data appears twice in the revised manuscript (as Table 2 and Ext Table 3)

Response: In response to the reviewer's suggestion in the first review, the summary of the X-ray data in Table 2 was moved to Supplementary Data (Extended Data Table 3). However, I forgot to remove Table 2. It has been removed in this revised manuscript. I appreciate the reviewer for finding this mistake.

- it would be good if the supplementary figures for review included in the rebuttal are actually included as extended data published with the manuscript.

Response: Following the suggestion by Reviewer #2, we have included Supplementary Figures for Review as Extended Data (Figs. 4 and 5) in the revised manuscript.

Response to Reviewer #2:

I greatly appreciate the responses by the authors to the comments raised. All my concerns have been addressed.

Response: We sincerely thank the reviewer for a positive evaluation.